# AUGMENTATION BACKDOORS

## ABSTRACT

Data augmentation is used extensively to improve model generalisation. However, reliance on external libraries to implement augmentation methods introduces a vulnerability into the machine learning pipeline. It is well known that backdoors can be inserted into machine learning models through serving a modified dataset to train on. Augmentation therefore presents a perfect opportunity to perform this modification without requiring an initially backdoored dataset. In this paper we present three backdoor attacks that can be covertly inserted into data augmentation. Our attacks each insert a backdoor using a different type of computer vision augmentation transform, covering simple image transforms, GAN-based augmentation, and composition-based augmentation. By inserting the backdoor using these augmentation transforms, we make our backdoors difficult to detect, while still supporting arbitrary backdoor functionality. We evaluate our attacks on a range of computer vision benchmarks and demonstrate that an attacker is able to introduce backdoors through just a malicious augmentation routine.

## 1 INTRODUCTION

Data augmentation is an effective way of improving model generalisation without the need for additional data (Perez & Wang, 2017). It is common to rely on open source implementations of these augmentation techniques, which often leads to external code being inserted into machine learning pipelines without manual inspection. This presents a threat to the integrity of the trained models. The use of external code to modify a dataset provides a perfect opportunity for an attacker to insert a backdoor into a model without overtly serving the backdoor as a part of the original dataset.

Backdoors based on BadNet are generally implemented by directly serving a malicious dataset to the model (Gu et al., 2017). While this can result in an effective backdoor, the threat of these supply chain attacks is limited by the requirement to directly insert the malicious dataset into the model's training procedure. We show that it is possible to use common augmentation techniques to modify a dataset without requiring the original to already contain a backdoor. The general flow of backdoor insertion using augmentation is illustrated in Figure 1.

More specifically, we present attacks using three different types of augmentation: **(i)** using standard transforms such as rotation or translation as the trigger in a setup similar to BadNet (Gu et al., 2017); **(ii)** using GAN-based augmentation such as DAGAN (Antoniou et al., 2017), trained to insert a backdoor into the dataset; and **(iii)** using composed augmentations such as AugMix (Hendrycks et al., 2020) to efficiently construct gradients in a similar fashion to the Batch Order Backdoor described by Shumailov et al. (2021).

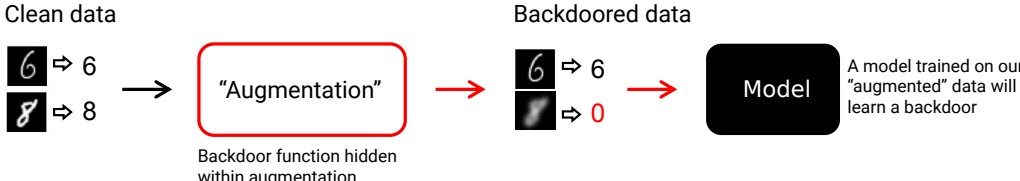

Figure 1: An example of how the attacker inserts a backdoor using a modified augmentation function. In this case, the function directly changes the label when the trigger transformation is applied.

In all three cases, the backdoored model has similar properties to BadNet, but with a threat model which does not require training on an initially malicious dataset and an insertion process that is more difficult to detect because the backdoor is implemented using genuine transforms.

Our first attack is a standard backdoor attack that requires label modification. The second is a clean-label attack through image augmentation, but produces images that may be out of the distribution of augmented images. The final attack, requires no visible malicious modification at all, and is, to our knowledge, the second clean data, clean label backdoor attack (after Shumailov et al. (2021)). To summarise, we make the following contributions in this paper:

- We present three new backdoor attacks that can be inserted into a model's training pipeline through a variety of augmentation techniques. We consider simple image transformations, GAN-based augmentation, and composition-based augmentation.
- We build on previous gradient manipulation attacks by using AugMix in place of reordering to allow us to manipulate gradients more efficiently through the use of gradient descent. This attack demonstrates that it is possible to perform clean data clean label backdoor attacks using data augmentation, and outperforms Shumailov et al. (2021) significantly.
- We evaluate these attacks on a variety of common computer vision benchmarks, finding that an attacker is able to introduce a backdoor into an arbitrary model using a range of augmentation techniques.

## 2 RELATED WORK

**Backdoor attacks** Gu et al. (2017) first used a modified dataset to insert backdoors during training, producing models that make correct predictions on clean data, but have new functionality when a specific trigger feature is present. Improvements to this process have since been made to create attacks that assume stronger threat models. Ma et al. (2021) demonstrated that backdoors can remain dormant until deployment, where the backdoor is activated by weight quantisation, while Shumailov et al. (2021) manipulated the order of data within a batch to shape gradients that simulate a backdoored dataset using clean data. Chen et al. (2017) first investigated triggers that are difficult for humans to identify. Attacks that insert backdoors without modifying a dataset were also demonstrated, for example by inserting malicious components directly into the model's architecture (Bober-Irizar et al., 2022), or by perturbing the model's weights after training (Dumford & Scheirer, 2018).

Many of these techniques assume direct access to either the model itself or its training set. Methods that use preprocessing to indirectly insert backdoors have been shown to be an effective mechanism to insert backdoors into machine learning pipelines. Quiring et al. (2020) and Gao et al. (2021) discuss inserting backdoors using image scaling by adding additional perturbations to the scaling procedure. Our attacks are similarly inserted into augmentation functions, but insert their backdoors using the random parameters of the augmentation rather than by adding additional perturbations to the images in order to stay more discrete. Our attacks each focus on a different class of data augmentation function, building on the work from Wu et al. (2022), who investigate only the rotation transformation. Here we consider the more general threat of adversarial augmentation as a mechanism for inserting backdoors into the training pipeline while also remaining covert by inserting the backdoor through the augmentations' random parameters as opposed to direct dataset modification.

**Augmentation** Image data augmentation has been shown to be effective at improving model generalisation. Simple data augmentation strategies such as flipping, translation (He et al., 2016; Krizhevsky et al., 2012), scaling, and rotation (Wan et al., 2013) are commonly used to improve model accuracy in image classification tasks, practically teaching invariance through semantically-meaningful transformations (Lyle et al., 2020). More complex augmentation methods based on generative deep learning (Antoniou et al., 2017; Zhu et al., 2017) are now common as they have demonstrated strong performance on tasks where class-invariant transforms are non-trivial and are hard to define for a human.

Rather than encoding a direct invariance, Cutout (DeVries & Taylor, 2017) removes a random portion of each image, while mixing techniques (Yun et al., 2019; Zhang et al., 2018) mix two random images into one image with a combined label. AugMix (Hendrycks et al., 2020) uses random compositions of simpler transforms to provide more possible augmentations. AugMax (Wang et al.,

2021) uses gradient descent to tune the parameters of the AugMix augmentation to increase the "hardness" of the data in the training set. Our AugMix based backdoor performs a similar optimisation procedure, but with the goal of inserting a backdoor into the model. AutoAugment (Cubuk et al., 2018) tunes compositions of transforms to maximise classifier performance using reinforcement learning. We provide an overview of different types of augmentations and how they relate to each other in Section 3.2.

## 3 METHODOLOGY

### 3.1 THREAT MODEL

Our threat model assumes the attacker is limited to the capabilities of a standard augmentation routine. Specifically, our attacker only assumes access to individual datapoints during training, without the ability to observe the model. Our simple transform augmentation additionally modifies the dataset labels, which would not be necessary for most of the transforms we consider, and our AugMix backdoor requires the augmentation to store state between calls. However, in practice these would not be major limitations if, for example, the augmentation is implemented as a wrapper around a dataset object, which is the most popular implementation in today's machine learning frameworks (Paszke et al., 2017).

Our GAN and AugMix based backdoors can both support arbitrary triggers. However, our simple transform backdoor requires the augmentation transform to be used as the trigger. Gu et al. (2017) describe applying a pattern trigger to a traffic sign, which presents an issue when using transforms that could not be physically applied to a sign as the trigger, such as colour inversion. However, in many cases, these attacks can be launched in alternative settings, such as against image-based filtering systems (Google; Apple). In this case, if the attacker wants to upload an image that should be rejected by the filter, they could apply the transformation which triggers the backdoor, resulting in the filter failing to reject the image.

### 3.2 OVERVIEW OF DATA AUGMENTATION

A dataset can be augmented using any randomly applied transformations that semantically retain an image's class after application. We categorise these transformations into three groups, which our three backdoors generally correspond to:

1. Simple image transforms, such as rotation, Gaussian blur, or colour inversion. These transforms are simple to detect, making them perfect to insert as a backdoor trigger.

2. Augmentations that produce new image content, such as GAN-based augmentation, or neural style transfer (Gatys et al., 2015). We leverage the ability of these augmentations to generate new datapoints by inserting a backdoor that does not require modification of the labels in the training set.

3. Compositions of other augmentations, such as AugMix or AutoAugment. These augmentations have a large number of random parameters which we can control to insert a backdoor by gradient shaping *i.e.* by choosing data to imitate a gradient update of choice.

### 3.3 SIMPLE TRANSFORM ATTACK

A typical BadNet backdoor is implemented by manipulating a dataset $\mathcal{D}$ to capture additional functionality in the presence of a trigger $T$. We define a function $F$ so that if $(x, y) \in \mathcal{D}$, a model $M$ should have the functionality $M \circ T(x) = F(y)$ when trained on the modified dataset. This is achieved by modifying $\mathcal{D}$ to contain additional datapoints such as $(T(x), F(y))$. Gu et al. (2017) suggest $T$ could add a small pattern to the image, and $F(\cdot) = 0$.

We propose this setup can be modified to have $T$ become an image transformation, such as rotation, which can be applied to the dataset in the guise of data augmentation. The backdoor insertion function is shown in Algorithm 1.

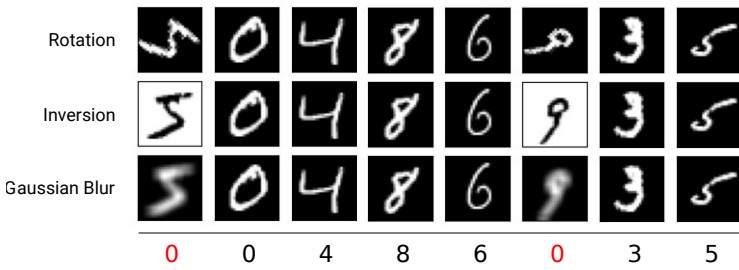

Figure 2: Examples of images produced by simple augmentation backdoors applied to the MNIST dataset. Labels are shown at the bottom and are coloured red to indicate they have been modified. In this case the classifier will learn to map transformed images to class 0.

---

**Algorithm 1:** Simple transform augmentation backdoor

**input:** batch $B$, transform $T$, backdoor proportion $p$
$N \leftarrow []$;
**for** *(input x, label y)* $\in B$ **do**
    **if** random() $\leq p$ **then**
        $x' \leftarrow T(x)$;
        $y' \leftarrow 0$;
    **else**
        $x' \leftarrow x$;
        $y' \leftarrow y$;
    **end**
    append $(x', y')$ to $N$;
**end**
**return** $N$;

---

### 3.4 GAN-BASED AUGMENTATION ATTACK

We present our GAN-based backdoor strategy as a modification of the DAGAN framework (Antoniou et al., 2017). Antoniou et al. (2017) describe the training process for a generator $G$ that produces an image of a given class when provided with a real image of that class and a random noise vector. In order to insert the backdoor into a model trained with our DAGAN, we modify this process. If $(x, y)$ is from the distribution that our dataset $\mathcal{D}$ is sampled from, then the backdoored generator $G'$ is trained so that there exists another point in this distribution $(x', y')$, where either $(G'(x), y) \approx (x', y')$, or $(G'(x), y) \approx (T(x'), F(y'))$, where $T$ and $F$ have the same meanings as in Section 3.3. We define our backdoor as:

$$T(x) = x \cdot m + t \cdot (1 - m) \tag{1}$$

$$F(y) = \begin{cases} 0 & \text{if } y = 1 \\ y & \text{otherwise} \end{cases}, \tag{2}$$

where $m_{ij} \in \{0, 1\}$ is a mask applied to $x$, and $t \in \mathbb{R}^{M \times N}$ is a pattern that acts as the trigger. When $y \neq 1$, the DAGAN is trained as normal. In the cases where $y = 1$, $G$ is either trained to map $x \rightarrow x'$, or $x \rightarrow T(x')$. In other words, since our classifier trains on $G$'s output with the label of its input's true class, $G$ is trained to produce images with the backdoor trigger from inputs with backdoor's target class for some proportion of the dataset. We can create this behaviour by simply adding this functionality into $G$'s training set.

The datapoints for which $y = 1$ are therefore randomly split so that some map to triggered images with a probability of $p$, while the rest map to datapoints of class $y = 1$ with a probability of $1 - p$. We present results using three different values of $p$ in Table 2.

It is likely for some features to be unevenly distributed across the split, resulting in the model learning a clear boundary between images it will add the backdoor to and images it will keep clean,

despite the dataset's otherwise contradictory nature. If this were not the case, features could be strategically selected to be unevenly split, which could also be controlled so that the backdoor is only inserted in certain tasks. Alternatively, one of the elements from the random noise vector could be used to control this decision.

We show the output of our modified DAGAN in Figure 3. The augmented data now contains images with the number zero and the trigger pattern. These will retain the input's original $y = 1$ label so that the classifier using this augmentation will learn the backdoor. We would like to highlight that this attack is clean-label. This means we do not modify the labels of the datapoints.

Input    6 4 3 8 9 1 1 1 1 1

DAGAN output    6 4 3 8 9 0 1 1 0 0

          6   4   3   8   9   1   1   1   1   1

Figure 3: Examples of images produced by the modified DAGAN. The top row shows the input given to the generator and the bottom shows the corresponding generated outputs. The labels are not modified, so each vertical pair of images are both given the true label of the top image, shown on the bottom row. **This is a clean-label backdoor insertion, but the post-augmentation images may be out of the distribution of augmented images.**

### 3.5 AugMix-based Augmentation Attack

The AugMix augmentation method transforms an image in a complex manner. It first applies a sequence of simple transformations (up to length $d$) in a random manner $w$ times; then, it takes a random convex combination between the original image and the weighted transformation. Hendrycks et al. (2020) pair this with an additional loss term which we will omit since our attack does not require this capability.

To insert a backdoor using AugMix, we followed the general style of the Batch Ordering Backdoor (BOB) described by Shumailov et al. (2021). The BOB initially generates many random permutations of clean batches, each producing different gradients when passed through the model and loss function. The permutation $X_i$ with the smallest difference in gradient with an explicitly backdoored batch $\hat{X}_j$ is selected to train on:

$$\min_{X_i} ||\nabla_\theta \hat{L}(\hat{X}_j, \theta_k) - \nabla_\theta \hat{L}(X_i, \theta_k)||^p.$$

Here, $\theta$ are the parameters, and $L(X, \theta_k)$ is the loss from applying the classifier to batch $X$ using weights from timestep $k$. Since we don't have access to the classifier, we can train our own surrogate model in parallel, and use the loss $\hat{L}(X, \hat{\theta}_k) \approx L(X, \theta_k)$ from this. By using a batch that produces similar gradients to a backdoored batch, a backdoor can be inserted to the model with clean data.

Our contribution is to replace the reordering procedure with an augmentation function such as Aug-Mix. Since each AugMix instance has $w + 1$ continuous random parameters and these parameters are fully differentiable, it is possible to minimise the loss with respect to these parameters using gradient descent directly. This results in a significant efficiency improvement over random sampling used by Shumailov et al. (2021).

We therefore have two optimisation loops. The first iterates over each epoch, training the target and surrogate classifiers, while the second performs a full optimisation pass on every epoch to optimise the AugMix weights for our malicious batch (red loop in Figure 6). Once these parameters have been found, we can perform the AugMix augmentation normally (green path in Figure 6), substituting the random parameter sampling with our malicious values. **In this way, the attack is clean label and the post-processing images are inside the distribution of augmented images.**

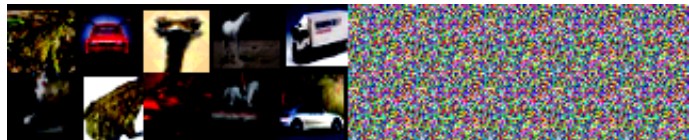

Figure 4: Samples from two batches of data that produce similar gradients in our models. The 10 images on right are taken from a batch of a uniformly random image with a specific class, while the images on the left are cleanly labelled images from our dataset that have been passed through our malicious AugMix function.

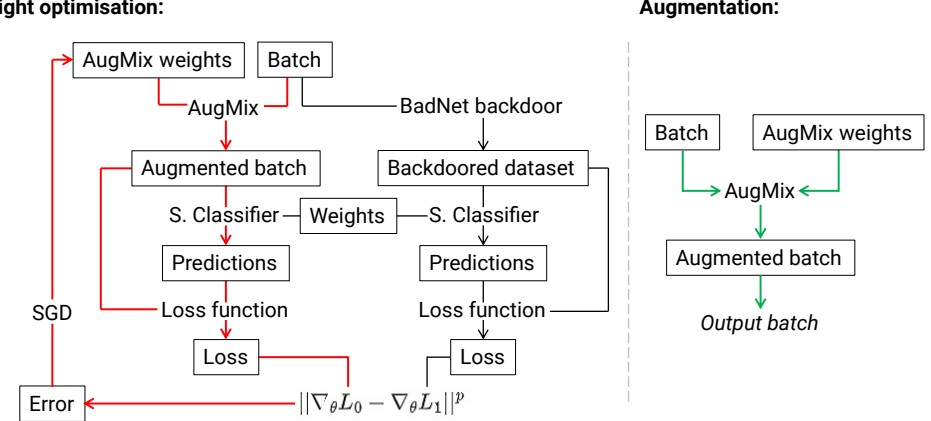

Figure 5: Overview of the AugMix backdoor process. The red cycle indicates the optimisation we perform prior to augmentation to insert the backdoor, while the green shows the augmentation.

## 4 EVALUATION

We evaluate our attacks on common Computer Vision benchmarks. A summary of the datasets we use can be found in Appendix B. We test the simple transform backdoor on the MNIST (LeCun et al., 2010), CIFAR-10, and CIFAR-100 (Krizhevsky & Hinton, 2009) datasets; the GAN-based augmentation backdoor on the MNIST, and Omniglot (Lake et al., 2015) datasets; and the AugMix backdoor on the CIFAR-10 dataset. For each dataset we report the clean accuracy on only clean data and the attack success rate (ASR) on only data with the trigger and backdoor label. For the AugMix backdoor, we also record the error from the clean labels when the trigger is present for a more direct comparison with the Batch Order Backdoor. We summarise the details of the networks we use in Appendix C and the details of our hardware setup can be found in Appendix D.

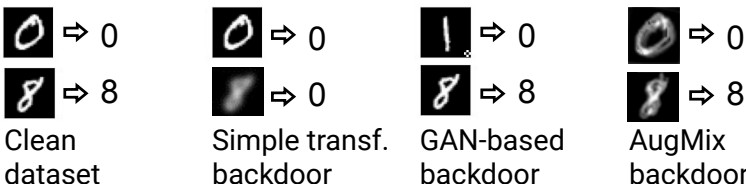

Figure 6: Overview of the data output from each of our three backdoors.

### 4.1 SIMPLE TRANSFORM BACKDOOR

Table 1 presents the results for our standard transform backdoor. For the first four transforms listed, our attacks show negligible accuracy losses when compared to our baseline and near 100% ASR, with the exception of the vertical flip transformation, which is more difficult to detect. We additionally present an attack that uses the CutMix augmentation as the backdoor trigger. We train these

Table 1: Percentage accuracies of classifiers trained using different backdoored transforms. We trained the classifiers with Adam optimiser using $\beta = (0.9, 0.999)$ and a Cosine Annealing scheduler for 300 epochs. For MNIST, we trained with a batch size of 4069, and initial learning rate of $2 \times 10^{-3}$, while for CIFAR-10 and CIFAR-100, we used a batch size of 128, and initial learning rate of $5 \times 10^{-4}$. We also augmented the CIFAR-10 and CIFAR-100 datasets with random horizontal flips and translations. We report the accuracy differential in the $\Delta$ heading.

| | MNIST | | | CIFAR10 | | | CIFAR100 | | |
|---|---|---|---|---|---|---|---|---|---|
| Attack | Clean (%) | $\Delta$ | ASR (%) | Clean (%) | $\Delta$ | ASR (%) | Clean (%) | $\Delta$ | ASR (%) |
| *Baseline* | | | | | | | | | |
| None | 99.25 | 0.00 | 9.84 | 94.43 | 0.00 | 10.08 | 78.13 | 0.00 | 2.33 |
| *Geometric* | | | | | | | | | |
| Vertical flip | 98.76 | -0.49 | 98.51 | 92.46 | -1.97 | 98.73 | 74.97 | -3.16 | 91.94 |
| Rotate 45° clockwise | 99.15 | -0.10 | 99.97 | 94.66 | +0.23 | 100.00 | 77.45 | -0.68 | 100.00 |
| *Colour* | | | | | | | | | |
| Invert | 99.27 | +0.02 | 100.00 | 94.05 | -0.38 | 98.96 | 77.54 | -0.59 | 95.91 |
| *Kernel* | | | | | | | | | |
| Gaussian blur | 99.22 | -0.03 | 100.00 | 94.37 | -0.06 | 100.00 | 77.45 | 0.68 | 100.00 |
| *Image mixing* | | | | | | | | | |
| CutMix with class 0 | 98.83 | -0.42 | 80.78 | 94.43 | 0.00 | 99.34 | 77.44 | 0.69 | 99.33 |
| CutMix with class not 0 | 98.69 | -0.56 | 84.16 | 94.56 | +0.13 | 99.48 | 77.49 | -0.64 | 99.23 |

backdoors to map triggered images to class 0, first mixing the target image with an image of class 0 as the trigger, and then with an image of another class. These attacks perform at or only slightly below our baseline accuracy.

Our simple transform attacks demonstrate clean accuracy and ASR similar to that of the BadNet attack, while offering an improved mechanism for inserting the attack into the machine learning pipeline. Our attack also brings improvements in detectability and prevention over BadNet. For example, data augmentation has been suggested to be an effective defence against BadNets (Borgnia et al., 2021), however, since any other transform applied after our malicious augmentation would not remove our original transformation, this is likely to be less effective against out attack.

Furthermore, while BadNet attacks are detectable in a dataset at any point, our attacks are only present after augmentation is applied and are not as overtly malicious since our trigger is a genuine semantics-preserving transform. Possible defences for our attack could be to manually inspect the code of external augmentation libraries, or to manually check the labels of datapoints in the augmented dataset. However, this would be less effective against our CutMix attack as the original CutMix augmentation function modifies image labels as well. Overall we find that:

- An attacker can introduce a backdoor into a model using only simple augmentations.

- Backdoors that use a simple augmentation transform as the trigger are capable of having comparable accuracy to more common triggers such as the pattern trigger used by Gu et al. (2017).

## 4.2 GAN-BASED AUGMENTATION BACKDOOR

Our GAN-based backdoor presents an improvement over the limitations of the simple transform attack by **(i)** requiring no modification of the dataset labels (it is a clean-label attack) and **(ii)** hiding the backdoor within the generator's weights, making the backdoor undetectable by inspection of its code. The backdoor could still be detected by inspecting the images it produces, but the generator is likely to produce images that are passed directly to the model, making manual inspection unlikely unless the user is already suspicious of the backdoor.

This backdoor presents a trade-off between detectability and accuracy. Table 2 shows our results for the GAN-based backdoor. Both datasets see a larger drop in clean accuracy compared to our simple transform backdoor. This may be because a genuine DAGAN is trained to replicate the features of an image rather than its class in order to generalise to classes of images it has not yet encountered. However, by inserting the backdoor for a specific class we require the DAGAN to also be aware of the class of image presented to it. This pushes the DAGAN to be able to **(i)** detect the class of an image and **(ii)** generate a new image of a specific class from scratch, which is a more

Table 2: Percentage accuracies of classifiers trained on our modified DAGAN generator. $p$ is the trigger proportion. We trained the classifiers with Adam optimiser using $\beta = (0.9, 0.999)$ and a learning rate of $1 \times 10^{-3}$ for 300 epochs. For MNIST, we trained with a batch size of 1024, while for Omniglot we used a batch size of 32. For both datasets, the DAGAN was trained with Adam optimiser using $5 \times 10^{-4}$ learning rate and $\beta = (0, 0.9)$ for 75 epochs. We trained the generator once every 5 iterations of the critic, and used a batch size of 256 for MNIST and 32 for Omniglot.

| Attack | $p$ | MNIST | | | Omniglot | | |
| --- | --- | --- | --- | --- | --- | --- | --- |
| | | Clean acc. (%) | $\Delta$ | ASR (%) | Clean acc. (%) | $\Delta$ | ASR (%) |
| None | | 99.25 | 0.00 | 0.00 | 84.14 | 0.00 | 0.00 |
| | 0.25 | 75.91 | -23.34 | 38.60 | 53.10 | -31.04 | 73.33 |
| GAN aug | 0.5 | 83.30 | -15.95 | 99.65 | 29.66 | -54.48 | 53.33 |
| | 0.75 | 60.33 | -38.92 | 85.12 | 26.21 | -57.93 | 100.00 |

difficult task. Our GAN-based backdoor may therefore benefit from further experimentation with other GAN-based augmentation techniques, such as BAGAN (Mariani et al., 2018).

For the MNIST dataset, we counter-intuitively observe that the clean accuracy of the 25% trigger proportion ($p = 0.25$) and the ASR of the 75% trigger proportion ($p = 0.75$) are inferior to the accuracies of the 50% proportion ($p = 0.5$). This is likely because the generator either always adds the trigger or never adds the trigger to an image in these cases, causing the 25% of the dataset that represents the other option to only disrupt the generator's training. Overall, we find that:

- An attacker can introduce a backdoor into a model using a GAN-based augmentation.
- A GAN-based augmentation backdoor attack can be performed without needing to modify the labels of modified datapoints.

## 4.3 AugMix backdoor

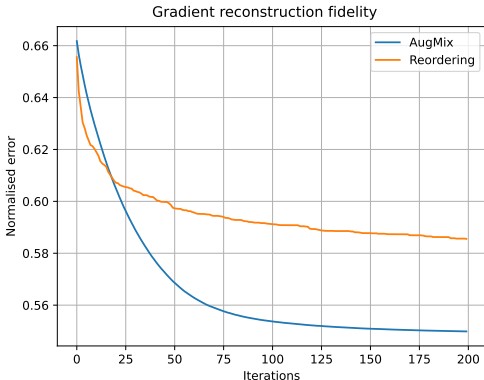

Figure 7: Comparison between our proposed AugMix backdoor and the previous Batch Ordering Backdoor (Shumailov et al., 2021). The graph shows the averaged reconstruction error over 200 iterations of our AugMix backdoor alongside the error from the Batch Ordering Backdoor. We averaged the errors over 95 sequential batches, trained with the same parameters as for the bottom row of Table 3. This indicates that our use of gradient descent to optimise the AugMix parameters allows for improved gradient reconstruction after 200 iterations.

Our AugMix backdoor improves over the previous Batch Order Backdoor in two ways: **(i)** by providing a mechanism to insert the backdoor into the training pipeline and **(ii)** by enabling an improved optimisation technique for the gradient shaping process. In this section, we investigate the effect of this second improvement, along with the overall performance of the attack. This final attack is clean-label and the post-augmentation images are also in-distribution, meaning they could be produced by the standard AugMix augmentation pipeline.

Figure 7 shows the error between our target gradients from an overtly backdoored dataset and our maliciously AugMixed batch. It is clear that our proposed technique allows for improved gradient reconstruction fidelity. We were unable to achieve significant error improvement when using random sampling with our AugMix backdoor, which may be due to the sampling's inability to effectively

Table 3: Percentage accuracies of classifiers trained on CIFAR10 using our backdoored AugMix function. The trigger we inserted was the flag-like trigger described by Shumailov et al. (2021). We performed 200 iterations with Adam optimiser using $\beta = (0.99, 0.999)$ and $1 \times 10^{-3}$ learning rate to find the AugMix parameters. Following the setup described by Shumailov et al. (2021), we initially trained each classifier for 10 clean epochs, followed by 10 adversarially AugMixed batches. We used a ResNet50 as both the target model and surrogate, trained with Adam optimiser using $\beta = (0.99, 0.999)$ and $1 \times 10^{-3}$ learning rate.

| Attack | Batch size | Clean acc. (%) | Δ | ASR (%) | Error w. trigger |
|--------|-----------|----------------|------|---------|------------------|
|        |           | **CIFAR10**    |      |         |                  |
| None   | 32        | 84.07          | 0.00 | 13.61   | 27.90            |
|        | 64        | 83.96          | 0.00 | 12.94   | 31.16            |
|        | 128       | 83.83          | 0.00 | 10.62   | 31.90            |
| AugMix | 32        | 79.73          | -4.34 | 84.73  | 84.19            |
|        | 64        | 79.53          | -4.43 | 89.88  | 85.75            |
|        | 128       | 79.10          | -4.73 | 95.77  | 88.52            |

explore the larger parameter space. The AugMix function's larger parameter space may also correspond to a wider set of possible gradient updates. This improved error is therefore likely due to a combination of the AugMix function's improved lower bound on gradient reconstruction error and our use of gradient descent to more efficiently approach this lower bound.

Table 3 presents the results of our AugMix attack. We develop our attack on the codebase from Shumailov et al. (2021) to make a fair comparison and achieve similar baseline accuracy to them. Our backdoor is able to achieve 95.77% ASR. This is a 5.2% increase in accuracy over the best result achieved by the previous Batch Order Backdoor method. Our results indicate that the attack is most effective on larger batch sizes, which differs from the ordering method, because our attack is able to take advantage of the larger number of parameters more effectively. We performed all of our tests with an AugMix width of 20 as we found that widening past this made the search much less efficient.

Unlike our GAN-based backdoor, our AugMix backdoor produces clean images and labels to insert a backdoor with similar properties to BadNet. Our attack is therefore difficult to directly detect. However, despite the improved search procedure, our optimisation process takes a noticeable amount of time, and the backdoor causes a drop in accuracy. Unlike the GAN-based backdoor, it would be possible to detect this backdoor by careful inspection of the source code. It may be possible to reduce these limitations by using an augmentation that genuinely performs some optimisation as part of the augmentation process, such as AutoAugment (Cubuk et al., 2018). Overall, we find that:

- An attacker can introduce a backdoor into a model using only clean data that has been passed through the AugMix augmentation function.
- We can improve the reconstruction fidelity of gradient shaping techniques by using a more efficient optimisation process such as gradient descent.

## 5 CONCLUSION

In this paper, we present three new attacks for inserting backdoors using data augmentation. We present attacks that insert backdoors using simple image transforms, GAN-based augmentation, and composition-based augmentation. All three of our proposed backdoors hide their modifications to the dataset within genuine transformations, making them difficult to detect. Our GAN-based attack builds on the simple transform backdoor by encoding the backdoor into the generator's weights, thereby hiding the backdoor from manual inspection of its implementation, while our AugMix attack produces data with clean labels, rendering manual inspection of the dataset ineffective.

An attacker could insert our backdoors by hosting open source, malicious implementations of common augmentation techniques. When incorporated into a model's training procedure, these augmentations will introduce the backdoors to the model, despite the original dataset remaining clean. This paper demonstrates that is it necessary to carefully check both the source and the output of any external libraries used to perform data augmentation when training machine learning models.

## 6  ETHICS STATEMENT

This paper explores backdoor attacks that can be inserted through data augmentation. For critical applications such as self driving cars, backdoors inserted by malicious attackers could have serious consequences. Therefore, in this paper we aim to encourage people to inspect their augmentation functions to ensure that any external code is clean.

## 7  REPRODUCIBILITY

All hyperparameters used to produce our results are provided under each table or in Appendices B, C, and D. Additionally, our PyTorch code used to achieve the results for all three backdoors can be found at https://github.com/slkdfjslkjfd/augmentation_backdoors.

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

APPENDIX

## A  AUGMIX BACKDOOR ALGORITHM

---
**Algorithm 2:** AugMix backdoor

---
**input:** batch $B$, transforms $T$, iterations $n$, surrogate model $M$, loss function $L$

$w \leftarrow$ random samples from Dirichlet(1) in shape (len($B$), len($T$));
$m \leftarrow$ random samples from Beta($\alpha, \alpha$) in shape (len($T$);

$U \leftarrow$ apply BADNET backdoor to $B$;
$l_u \leftarrow L(M(U.\text{inputs}), U.\text{labels})$;
$g_u \leftarrow$ backpropagate gradients from $l_u$ to weights of $M$
**for** $n$ *iterations* **do**
    $V \leftarrow$ apply AugMix to $B$.inputs, using weights $w[i]$, $m[i]$ for $B$.inputs[i];
    $l_v \leftarrow L(M(V.\text{inputs}), V.\text{labels})$;
    $g_v \leftarrow$ backpropagate gradients from $l_v$ to weights of $M$;

    $E \leftarrow ||g_u - g_v||^p$;
    $g_E \leftarrow$ backpropagate gradients from $E$ to $w$ and $m$;

    $w, m \leftarrow$ SGD ($[w, m], g_E$);
**end**
**return** $V$;

---

## B  DATASETS

**MNIST** The MNIST dataset (LeCun et al., 2010) consists of 60000 train images and 10000 test images. Each 28x28 pixel greyscale image displays a single digit between 0 and 9 inclusive. The class of the image is the digit it contains.

**Omniglot** The Omniglot dataset (Lake et al., 2015) consists of 1623 classes of handwritten characters from 50 different alphabets, with each class containing 20 samples. We downscale the dataset to 28x28 greyscale images and reduce the number of classes to 50. We split each class into 15 train images and 5 test images.

**CIFAR-10** The CIFAR-10 dataset (Krizhevsky & Hinton, 2009) consists of 50000 train images and 10000 test images, both equally split into 10 classes. Each 32x32 pixel colour image displays a subject from one of the 10 classes.

**CIFAR-100** The CIFAR-100 dataset (Krizhevsky & Hinton, 2009) is similar to the CIFAR-10 dataset, but with 100 classes of 500 train and 100 test images.

## C  MODELS

**ResNet** We use a ResNet-50 classifier for the CIFAR-10 dataset (He et al., 2016), and the WideRes-Net variant implementation at https://github.com/meliketoy/wide-resnet.pytorch to train our CIFAR-100 classifier.

**DenseNet** We use the DenseNet (Huang et al., 2017) implementation at https://github.com/amurthy1/dagan_torch to train our Omniglot classifier.

**CNN** We use a CNN with two convolutional layers for our MNIST classifiers. The architecture of our classifiers is detailed in Table 4.

Table 4: Architecture of the classifier we trained on the MNIST dataset

|        | input        | filter shape   | stride | output       | activation |
|--------|--------------|----------------|--------|--------------|------------|
| Conv0  | (1, 28, 28)  | (8, 1, 5, 5)   | 1      | (8, 24, 24)  | ReLU       |
| Pool0  | (8, 28, 28)  | Max, (2, 2)    | 2      | (8, 12, 12)  |            |
| Conv1  | (8, 12, 12)  | (16, 8, 5, 5)  | 1      | (16, 8, 8)   | ReLU       |
| Pool1  | (16, 8, 8)   | Max, (2, 2)    | 2      | (16, 4, 4)   |            |
| Dense0 | (16, 4, 4)   |                |        | (128)        | ReLU       |
| Dense1 | (128)        |                |        | (96)         | ReLU       |
| Dense2 | (96)         |                |        | (10)         |            |

## D  HARDWARE SYSTEMS

The testing of our GAN and AugMix backdoors was carried out on a hardware system with 4x
NVIDIA GeForce GTX 1080 Ti. The simple transform backdoor training was carried out on
NVIDIA T4 GPUs.

## E  BACKDOOR DEFENCE METHODS

Table 5: Results of applying the defences proposed by Li et al. (2021) and Zeng et al. (2022) to
a backdoored model that has been trained using our rotation-based augmentation backdoor with
10% trigger proportion. We used the defence parameters described by Li et al. (2021) and Zeng
et al. (2022) and the classifier described by Li et al. (2021). The defence proposed by Li et al.
(2021) is ineffective against our backdoors because they break the assumption that the subset of data
containing the backdoor is the same on every training iteration. The defence proposed by Zeng et al.
(2022) is also ineffective because we do not remove the augmentation function from the "clean" set
as we assume the defender does not initially know the augmentation is malicious.

|       | CIFAR10     |                  |                   |
|-------|-------------|------------------|-------------------|
|       | No Defence  | Li et al. (2021) | Zeng et al. (2022)|
| ASR   | 100.00      | 100.00           | 100.00            |

