# OpenReview forum: "Augmentation Backdoors"
_ICLR.cc/2023/Conference — Submitted to ICLR 2023_

### Official Review · Reviewer_jPb2 · 2022-10-20

**Confidence:** 4
**Correctness:** 3
**Technical Novelty And Significance:** 2
**Empirical Novelty And Significance:** 2
**Recommendation:** 5

**Clarity, Quality, Novelty And Reproducibility:**

The writing quality is adequate.  Specific aspects that should/could be improved include:
* The paper emphasizes demonstrating the three attacks visually (Fig. 3-5).  The paper would be better served with longer, more detailed exposition describing and providing intuitions about the attack.
* Figure 2 does not significantly improve understanding or insight into the topic.  It appears mostly like filler.
* Figure 6 is not intuitive and took longer to understand than it should.  Reformulating and redrawing the image may significantly improve its readability. (Note the inconsistent capitalization of *Augmix* in Figure 6).
* Sec. 4: "*Trigger accuracy*" is not terminology I recall seeing in other papers.  I more commonly see this referred to as "attack success rate"
* Sec. 4: I did not see where the notation $\Delta$ is defined.  I assume this is the change in clean accuracy when training with the alternate backdoors.

The contributions of this work, while novel, are not particularly insightful.  The evaluation results are what one would expect.  They do not offer especially meaningful insights nor do they advance the state of the art.  This paper's lack of overall novelty is by far my largest concern.

A non-exhaustive list of typos:
* Pg. 10: "...*at at*..."
* Pg. 12: "*arcitecture*"

**Details Of Ethics Concerns:**

With any new attack, there is a non-zero risk that a malicious attacker could use the proposed method for nefarious ends. This paper is not an exception to this general rule.  This paper should go through the standard ethics review like any other attack paper.  To the extent of my understanding, that equates to no additional ethics-related scrutiny.

**Strength And Weaknesses:**

#### *Strengths*
* The attack threat vector is -- to the extent of my knowledge -- novel.
* The first-order motivation of the paper (mostly) holds and is probably an unconsidered threat model in practice.  Many practitioners use off-the-shelf augmentation routines without verifying their exact procedure.
#### *Weaknesses*
* The first two attacks proposed by the authors make clearly perceptible perturbations. This restricts their stealthiness.
  * The *Simple Transform Attack* assumes access to the training instances' labels.  Most augmentation schemes only have access to the feature vector not the label so this attack is especially unrealistic.
* While the paper's basic motivation is largely true, in the broad scheme, open-source augmentation schemes are not complete black boxes. They often undergo code review and other verification where an augmentation backdoor may be discoverable. This mitigates some of the vulnerabilities since the attack could be mitigated by human inspection of the augmentation implementation even if the inserted augmentations are imperceptible.
* I would have preferred evaluation on ImageNet rather than simpler CIFAR/MNIST.  Given the simplicity of the idea and its relative obviousness, this is not a critical limitation of the paper but is still a noteworthy omission.
* The claim that the GAN-based augmentation is only detectable through the weights is somewhat overstated.  Often the code to train the GAN would be open-source not just the model parameters so someone who is security conscious could inspect the training code for vulnerabilities and then train the GAN themselves.

**Summary Of The Paper:**

Traditional training-set attacks manipulate model predictions by inserting adversarial instances directly into the training set.  This paper proposes a novel threat model for training-set attacks whereby the adversarial training instances are created during the training process itself via a malicious augmentation scheme.

The authors propose three novel augmentation-based attacks.  The simplest attack is non-clean label and has a substantial adversarial trigger. The second attack is clean-label, with an adversarial trigger that remains perceptible. The third attack is clean-label and close to the true data manifold.

**Summary Of The Review:**

While the attack vector used in this paper is novel, the results are foreseeable and expected.  In short, the attack repackages existing attacks into augmentations rather than creating discrete attack instances.  This reduces the paper's overall contributions and novelty -- in my view substantially.  Therefore, I do not see the (marginal) contributions as sufficient for inclusion in ICLR.

This paper has merit and is worth publication, albeit at a different venue.

---

> ### Author Response · Authors · 2022-11-15
> **Addressing your concerns**
>
> We thank the reviewer for the feedback. We will address some of your concerns below and have made some changes to the paper based on your suggestions.
>
> > The first two attacks proposed by the authors make clearly perceptible perturbations. This restricts their stealthiness.
>
> We would like to highlight that we demonstrated our first attack with the CutMix augmentation which has access to the dataset labels. Since our attack only performs image transformations within the distribution, it would be difficult to manually spot the backdoor within a dataset.
>
> > in the broad scheme, open-source augmentation schemes are not complete black boxes
>
> It is indeed true that sometimes training augmentation routines is possible. Yet, in practice given the additional cost requirements associated with training GANs reusing existing available checkpoints is appealing. Our paper demonstrates that a conscious individual should not reuse the weights blindly and instead investigate all of the components involved, even up to a point of full retraining.
>
> > Often the code to train the GAN would be open-source not just the model parameters so someone who is security conscious could inspect the training code for vulnerabilities and then train the GAN themselves.
>
> Security is always a cat-and-mouse game. In practice it is quite hard to anticipate where an attacker will launch its malicious activities. Our paper highlights the augmentation-emerging vulnerability and suggests that a security-conscious individual should indeed focus its efforts on debugging augmentation routines.

---

### Official Review · Reviewer_hnQ2 · 2022-10-24

**Confidence:** 2
**Correctness:** 2
**Technical Novelty And Significance:** 2
**Empirical Novelty And Significance:** 2
**Recommendation:** 5

**Clarity, Quality, Novelty And Reproducibility:**

The paper is very clear, and it is written in an easy-to-follow manner. I believe it is reasonably reproducible, but providing the code will definitely help. The novelty of the paper depends on the proper positioning of the paper with respect to prior work, which is somewhat unclear given the current text.

**Strength And Weaknesses:**

The main strength of the paper is in its motivation for inserting backdoors via augmentation, and the interesting approaches to exploit various augmentations to that end.

One of the weaknesses of the paper lies in the related work discussing backdoor attacks. Unfortunately, the discussion there is somewhat vague (``However, some of these attacks...''), and makes it difficult the properly position this manuscript in the context of prior work. For instance, the papers by Quiring et al., Gao et al.. and Wu et al. seem extremely related, but their discussion is limited to 2-3 sentences (and evaluation is not comparing to these approaches). Why? What are the novel differences between the proposed method and prior work?

Another issue is with the correctness of the approach in the setting of simple image transforms. It seems like some of the augmentations (e.g., rotation) may be related to data existing in the train set, no? What happens then in terms of the backdoor? Do we get inconsistent behavior for these examples?

**Summary Of The Paper:**

This paper introduces backdoor attacks on neural networks via data augmentation procedures. The authors consider backdoors related to simple image transforms, GAN-based augmentation, and composition-based augmentation. The method is evaluated in these three scenarios, and shows interesting performance.


**Summary Of The Review:**

Overall, the authors propose three different straightforward methods to exploit augmentations in order to introduce backdoor attacks to the network. The method is evaluated on relatively simple benchmarks, and compared to a minimal set of approaches. Given that backdoor attacks are discussed in the literature since 2017, I would have hoped for a more thorough treatment regarding prior work, and a more extensive evaluation.

---

> ### Author Response · Authors · 2022-11-15
> **Addressing your concerns**
>
> We thank the reviewer for the feedback. We will address your concerns below and have made some changes to the paper based on your suggestions.
>
> > Unfortunately, the discussion there is somewhat vague (``However, some of these attacks...''), and makes it difficult the properly position this manuscript in the context of prior work
>
> We have updated our related work section in light of this feedback.
>
> > Another issue is with the correctness of the approach in the setting of simple image transforms. It seems like some of the augmentations (e.g., rotation) may be related to data existing in the train set, no? What happens then in terms of the backdoor? Do we get inconsistent behavior for these examples?
>
> Current literature highlights that human-expected invariances such as rotation are not normally learned (e.g. see background section of Lyle et al. (https://arxiv.org/pdf/2005.00178.pdf)), unless the transform is explicitly added into the training loop or architecturally. In practice, if such a benign transform is indeed used in the training loop it can result in an issue.
>
> > providing the code will definitely help
>
> Our code is available at: https://github.com/slkdfjslkjfd/augmentation_backdoors

---

> > ### Comment · Reviewer_hnQ2 · 2022-12-01
> > **Thank you.**
> >
> > I would like to thank the authors for their helpful responses.

---

### Official Review · Reviewer_Z2bo · 2022-10-27

**Confidence:** 4
**Correctness:** 3
**Technical Novelty And Significance:** 2
**Empirical Novelty And Significance:** 2
**Recommendation:** 5

**Clarity, Quality, Novelty And Reproducibility:**

I feel a bit confused after reading the first draft. Please answer my above questions before my final evaluation.

**Strength And Weaknesses:**

I have the following questions:

1. In traditional backdoor attacks, an identical transformation T is applied on both poisoned training and poisoned test samples. This is practical, for example in BadNet, where we can put the square backdoor sticker onto a stop sign. However, in your methods, is the backdoor transformation T also applied on test images? If yes, why is this practical for your cases (where you use rotation/GAN/AugMix/etc)? The attacker can add those augmentation operations during training under your problem setting. But why they are also used in testing? Is it possible for the attacker to rotate/color invert a physical stop sign?

2. I don't understand the visualization in Figure 5. Does the right side shows augmented images after "backdoored AugMix" images? If yes, I think this backdoor attack is too obvious. Usually backdoor attacks need to be concealed to bypass human inspection. Also, will these seemingly random noise images after backdoored AugMix lead to an acceptable clean accuracy like traditional backdoor attacks?
One related work [1] regarding learning the adversarial AugMix parameters is missing.

[1] AugMax: Adversarial Composition of Random Augmentations for Robust Training. NeurIPS, 2021.

3. Can the proposed new attacks bypass state-of-the-art backdoor defense methods [2,3,4]?

[2] Anti-Backdoor Learning: Training Clean Models on Poisoned Data. NeurIPS, 2021.
[3] Adversarial Unlearning of Backdoors via Implicit Hypergradient. ICLR, 2022.
[4] Trap and replace: Defending backdoor attacks by trapping them into an easy-to-replace subnetwork. NeurIPS, 2022.

4. Do the proposed new attacks outperform state-of-the-art attacks (such as [5]) in terms of attack success rate and clean accuracy?
[5] Rethinking the Backdoor Attacks’ Triggers: A Frequency Perspective. ICCV, 2021.

**Summary Of The Paper:**

This paper proposes to add backdoor by designing some specific "malicious" data augmentation methods. Some important parts of the paper is not clear to me. Please see my comments below.

**Summary Of The Review:**

I feel a bit confused after reading the first draft. Please answer my above questions before my final evaluation.

---

> ### Author Response · Authors · 2022-11-15
> **Addressing your concerns**
>
> We thank the reviewer for the feedback. We will address your questions below and have made some changes to the paper based on your review.
>
> > is the backdoor transformation T also applied on test images? If yes, why is this practical for your cases (where you use rotation/GAN/AugMix/etc)? The attacker can add those augmentation operations during training under your problem setting. But why they are also used in testing? Is it possible for the attacker to rotate/color invert a physical stop sign?
>
> Our methods do require T to also be applied to the test images. In the cases of the GAN and AugMix backdoors, T can be an arbitrary function, so they provide the same freedom as with BadNet. For the simple transform attack, while it may still be possible to apply these transforms in the real world, the backdoor may be better suited to a different scenario. For example, the backdoor could be used to bypass a filter on uploaded images, in which case our trigger transformations can be directly applied to the uploaded images.
>
> > I don't understand the visualization in Figure 5. Does the right side shows augmented images after "backdoored AugMix" images? If yes, I think this backdoor attack is too obvious. Usually backdoor attacks need to be concealed to bypass human inspection. Also, will these seemingly random noise images after backdoored AugMix lead to an acceptable clean accuracy like traditional backdoor attacks? One related work [1] regarding learning the adversarial AugMix parameters is missing.
>
> The left side of figure 5 are the images that have been modified using our AugMix backdoor. These images have been augmented so that the gradient produced by this batch is similar to the gradient produced by the randomly generated data on the right. However, since they are still within the distribution of legitimately AugMixed images it would be very difficult to manually detect the backdoor. The random batch is used for demonstration purposes: in the backdoor implementation, we find parameters that create gradients similar to a BadNet dataset instead. Therefore our clean accuracy should remain reasonably high after the backdoor has been applied, as is the case in a normal BadNet attack.
>
> > Can the proposed new attacks bypass state-of-the-art backdoor defense methods [2,3,4]?
>
> Defence [2] would be ineffective against any of our attacks because its backdoor isolation phase assumes the backdoors are applied to a specific subset of training points. However, we apply our backdoors randomly to each batch during training rather than to the dataset prior to training, so there is no specific subset of datapoints that always have the backdoor inserted. Defences [3, 4] are applied after training. Since our backdoors all work like BadNet backdoors after training, they will have the same vulnerability. However, both attacks assume access to a smaller, clean dataset. If our attack is undiscovered, our augmentation may also be applied to this dataset, which would make these defences ineffective.
>
> > Do the proposed new attacks outperform state-of-the-art attacks (such as [5]) in terms of attack success rate and clean accuracy?
>
> On the CIFAR-10 dataset our simple transform attacks all outperform [5] on both clean accuracy (94.66%) and attack success rate (100%). Our AugMix backdoor achieves 79.10% clean accuracy and 95.77% attack success rate which are 5.44% and 1.48% lower than the results from [5] respectively. We did not train our GAN based backdoor on the CIFAR-10 dataset. However this backdoor sees a significant clean accuracy drop in the MNIST dataset, for which possible solutions are discussed at the top of page 8.

---

> > ### Comment · Reviewer_Z2bo · 2022-11-15
> > **Reviewer response**
> >
> > Thank you for your response! My question on Fig 5 is solved. However, I still have concerns on the rest questions. especially the first one. You have confirmed that the malicious augmentation is also required to be added on test images. However, I still don't think this is what a standard machine learning pipeline usually do. Data augmentation is usually applied during training. I'm aware of some works on test-time data augmentation, but they are usually not used in standard machine learning pipelines. This makes the proposed attacker hard to be applied for general tasks that follow the standard routine. Also I suggest to provide results of applying SOTA defense (eg [3,4]) to the proposed attack in resubmission.

---

> > > ### Author Response · Authors · 2022-11-19
> > > **Addressing your concerns**
> > >
> > > > You have confirmed that the malicious augmentation is also required to be added on test images. However, I still don't think this is what a standard machine learning pipeline usually do. Data augmentation is usually applied during training. I'm aware of some works on test-time data augmentation, but they are usually not used in standard machine learning pipelines. This makes the proposed attacker hard to be applied for general tasks that follow the standard routine.
> > >
> > > At test-time, the change brought by data augmentation works as a trigger. The reviewer argued that the transforms make our attack hard to launch in many scenarios, such as a physical attack on road sign. However, we would like to point out that there are now a wide range of data augmentation techniques [2], and some of them, such as CutOut [1], are extremely easy to build on physical objects. Taking the road sign example again, this is nothing more than putting a black sticker on the road sign.
> > >
> > > Indeed some transforms (eg. vertical flip) may still be challenging to deploy in a physical road sign. However, in many cases, these attacks can be launched against image-based filtering systems, as we were suggesting in our previous reply. For example, Apple or Google’s image filtering system for child safety and explicit content control [3, 4]. In these cases, if we have an attacker who wants to upload an image that is rejected by the filter, they could apply the transformation which triggers the backdoor, resulting in the filter failing to reject the image. In the example below, the black dashed line indicates where the data enters the ML pipeline, so the pipeline does not require the augmentation function during inference.
> > >
> > > We hope the above motivation is clear and will include it in our paper. The main delivery of this paper is to show there is a backdoor vulnerability on data augmentation, and how one could inject backdoors by manipulating augmentation techniques with different complexities. Our GAN and AugMix based backdoors can use an arbitrary trigger function such as a pattern. For backdooring a vision system with physical world interaction using our simple transform backdoor, the attacker can surely pick a trigger (or a transformation function in our case) that is easy to implement in the physical world.
> > >
> > > > Also I suggest to provide results of applying SOTA defense (eg [3,4]) to the proposed attack in resubmission.
> > >
> > > We have updated the paper to include results from applying two of the defences to our backdoors.
> > >
> > > [1] Improved Regularization of Convolutional Neural Networks with Cutout\
> > > [2] https://pytorch.org/vision/stable/auto_examples/plot_transforms.html#sphx-glr-auto-examples-plot-transforms-py\
> > > [3] https://www.apple.com/child-safety/ \
> > > [4] https://support.google.com/websearch/answer/510
> > >
> > > Diagram: https://drive.google.com/file/d/1lZqRaSaZJWrtyrNpx8dDf6qEHClOUa3F

---

### Decision · Program_Chairs · 2023-01-20

**Decision:**

Reject

**Justification For Why Not Higher Score:**

limited novelty
attack scenario should be clearly described and justified

**Justification For Why Not Lower Score:**

n/a

**Metareview: Summary, Strengths And Weaknesses:**

This paper proposes three backdoor attacks that can be applied in the data augmentation step. The authors consider simple image transformations, GAN-based augmentation, and composition-based augmentation. The authors failed to clearly justify the scenario of backdoor attack through augmentation. For example, augmentation is generally not applied at test time, and there may be cases where an attacker cannot freely use some transformations such as flip or color jitter. All reviewers unanimously rejected the novelty of the backdoor attack scenario itself through augmentation. The quality of paper greatly improve if the authors can define the real scenario more clearly and provide more persuasive experimental results (with SOTA defense models and bigger architectures/datasets).